# Between Migrant Care Work and New Occupational Welfare Tools: Changing Home Care Arrangements in Italy

**DOI:** 10.3390/ijerph17155511

**Published:** 2020-07-30

**Authors:** Georgia Casanova, Mirko Di Rosa, Oliver Fisher, Giovanni Lamura

**Affiliations:** 1Center for Socio-Economic Research on Ageing, IRCCS-INRCA—National Institute of Health and Science on Ageing, 60121 Ancona, Italy; o.fisher@inrca.it (O.F.); g.lamura@inrca.it (G.L.); 2Unit of Geriatric Pharmacoepidemiology and Biostatistics, IRCCS-INRCA—National Institute of Health and Science on Ageing, 60121 Ancona, Italy; m.dirosa@inrca.it; 3Department of Economics and Social Sciences, Università Politecnica delle Marche, 60121 Ancona, Italy

**Keywords:** occupational welfare schemes, home care, social innovation, migrant care work, Italy

## Abstract

Austerity measures on services provision, introduced due to recent economic crises, have stimulated the search for innovative welfare solutions, including options that are not directly or entirely funded by public sources. In Italy, recent legislation has promoted the development of occupational welfare (OW) measures, aimed at strengthening the supply of services to support employees with informal (elder) care responsibilities. This paper aims to describe how the newly introduced OW schemes might innovate existing care arrangements, by identifying their impact on the different actors involved in home care provision (care recipients, family carers, home care service providers and migrant care workers), as well as at a macro level in terms of promoting social innovation. The international relevance of the Italian case comes from the fact that it is one of the more representative familistic care regimes, largely characterized by home care provided by informal carers and migrant care workers (MCW). The importance of Italian OW schemes is increasing, and in 2018 their presence in company-level bargaining agreements grew by more than 15%. A rapid review of the literature and expert interviews allowed us to describe the complex Italian OW schemes system, and to identify the positive implications of their application for the country’s long-term care (LTC) context, underlining what makes these measures a clear example of “social innovation” likely to have a future positive impact on home-based care in Italy.

## 1. Introduction

In the last decade, the international debate on home care provision for frail elders has repeatedly underlined the importance of recognizing the role of informal care within a partnership approach in care management, in order to achieve a more effective long-term care (LTC) system [1,2,3,4,5].

The challenges posed by the recent international economic crisis that started in 2008 and the consequent austerity measures imposed, among other things, on public expenditure for care services and social protection policies, has stimulated the search for alternative welfare solutions, including options that are not directly or entirely funded by public sources [6,7]. A not always explicit corollary of this approach is that citizens are today called upon to be increasingly responsible for managing their care arrangements, thus *de facto* to rely more heavily on care provided by relatives, friends, and neighbours [8,9,10]. This lies, on the one hand, on the continued demographic trends leading to an increase in the percentage of elders on the whole population, as in 2018 almost 20% of the European population was aged 65 or older, with an overall increase of more than 2 percentage points from 10 years earlier (Table 1). On the other hand, a remarkable share of the population is called to provide a response to the care demand emerging from these trends. In the same year, on average 5.5% of European people between the ages of 18 and 64 declared that, in their role as informal carers, they shoulder a specific responsibility to support incapacitated relatives (including child and elder care responsibility), and 4% of them stated that they provided care exclusively to enable (older) adult relatives’ activities of daily living (non-reported data).

In this regard it should be noted that, traditionally, the level of caring responsibility is particularly high in countries with familial care regimes (Italy, Spain, Portugal, Greece, Ireland and Malta) and transition care regimes (Latvia, Poland, Estonia, Croatia), because families are strongly involved in providing informal care [11,12]. Table 1, however, underlines that also in countries more oriented to a mixed care regime and inclined to involve private providers [13], people are extensively responsible for providing care to their relatives too (e.g., The Netherlands and Ireland).

Noticeably, the intensity of population ageing is particularly relevant in countries where the share of people with informal care responsibilities is already high. However, while in the familial care regime countries (e.g., Italy, Portugal and Greece) old age rates are among the highest, in other countries the main challenge is represented by the speed of the ageing process. This is true for example in Finland, the Netherlands, Malta and Czechia, where the ratio of the number of people aged 65 and over to those aged 0–14 is growing at more than double the speed of the European average. Since the family remains, with no exceptions, the provider of the bulk of long-term care provision, the impact of demographic ageing could therefore be very relevant on the large segments of the population involved in informal care. Already in 2016, an average of 17% of the European population declared that they provide care for their relatives weekly. Across all countries, women were more likely than men to declare their status as informal care providers. Table 1 highlights that at the EU level the rate of informal caregivers is at its highest among women and those in working age (i.e., between 35 and 64). Indeed, the international literature highlights that family carers are more likely to be women and part-time workers [14,15] and that, when no adequate support is provided, these more frequently experience income losses and, in the long run, a weaker position in the labor market [15,16]. Therefore, policies that support working carers and improve the balance between formal and informal care potentially offer a strategic solution to promoting social innovation in home care.

Within this international context, recent legislation in Italy has promoted the adoption of welfare measures at the company level (also referred to as occupational welfare schemes), aimed at strengthening the supply of services to support employees with informal (elder) care responsibilities. In this regard, Natali and Pavolini [7] modernized the concept of Occupational Welfare (OW) offered by Titmuss [17] and Greve [18], by defining OW “as the sum of benefits and services provided by social partners, employers and trade unions (by themselves or with the participation of others) to employees over and beyond the public benefits, based on an employment contract”.

Since the 1990s, OW policies have been spreading throughout Europe, although this has occurred in a still fragmented manner. In this respect, at least two main OW strategies can be identified: in northern European countries, OW is characterized by a highly formal welfare supply, so that OW works to support the existing offer of residential and home care provision; in France and southern European areas, instead, OW still has room for development to accommodate the growing care demand for employees and their families [6,19]. Within this geographical framework, the relevance of the Italian case is important for two reasons. Firstly, this innovative Italian scheme is applied in a notoriously “familistic” care regime, in which home-based care is provided jointly by family members and, increasingly, by privately paid (primarily migrant) care workers (MCWs), similarly to what has been witnessed by other Mediterranean and Central European countries [20,21]. Secondly, and with regard to the MCWs, Italy has been facing a shift in the reliance on this source of support: until 2008, regularly employed care workers were almost exclusively migrants (mainly from Eastern Europe), while in recent years the number of Italian care workers has steadily increased, so that today Italians are the second most common group of care workers after Eastern Europeans.

These developments are deeply changing the landscape of home-based LTC provision in several countries, but so far very little attention has been paid to the identification of policies able to govern these phenomena, which are to a large extent either invisible or tolerated, as they allow the current LTC systems to keep going at an apparently low cost for the public finances. In Italy, in particular, LTC provision involves multiple public and private (for-profit and non-profit) stakeholders, with different and often overlapping roles, which are defined by legislation. In this regard, the impact of private care providers in home care provision is still quite marginal and mostly limited to the sector of complementary or integrative insurances.

In light of these circumstances, this paper aims to investigate whether and how the recently introduced OW schemes in Italy might be effective in innovating existing care arrangements. This analysis will specifically attempt to identify their impact at different levels: (a) at a *micro level*, in terms of quality of care and of life concerning all main actors involved: older care recipients; informal carers; home care service providers; and migrant care workers; (b) at a *meso level*, through the analysis of the role played by the private sector (including enterprises stimulated by OW schemes) in promoting new initiatives to support employees in better tackling their family care challenges; (c) at a *macro level*, on the promotion of innovations in the LTC sector as a structural component at a system level. To contextualize the relevance of the Italian case for the European context, some policy recommendations will be also formulated. The aim is to provide decision makers and practitioners with suggestions on how to best implement OW policies in different areas of welfare, in order to achieve more integrative and innovative strategies to tackle both policy and services fragmentation. This is particularly urgent since, to our knowledge, no study has so far collected and reported comprehensive data on the results of this still only recently implemented scheme. By doing so, the paper hopes finally to meaningfully contribute to the current debate in the specialised literature on OW and home care issues, and to boost interest on a topic deserving more systematic studies for the future.

## 2. Materials and Methods

In order to achieve the aims illustrated above, this study combines data collected by means of a rapid literature review with expert interviews. This mixed approach has been deemed as the most appropriate to capture “in an agile manner” both the context-specific characteristics and the rapidly evolving impact of OW policies on the Italian LTC system.

As for the first component of this study, the rapid review method was considered as more suitable—compared to the certainly more comprehensive but also much lengthier alternative of carrying out a systematic review—to ensure swiftly available results on a topic on which, in Italy and elsewhere, there is urgency to deliver evidence to inform not only the academia, but also practitioners and policy makers. Following the principles underpinning this method [22], the first author extracted articles using Google, Google Scholar, and Scopus. The search terms used were: “Occupational welfare”, “Corporative welfare scheme”, “Long-Term Care”, “Europe” and “Italy”. The search identified in a first step a total of 182 English and Italian records, excluding duplicates. The web search allowed to reduce this number in case of records linked to more than one keyword, and to understand how the concepts of “Occupational Welfare” and “Corporate Welfare schemes” were used in the literature. All records have been screened by the first author by a double check. Firstly, the selection was based on title and abstract to understand their relevance and applicability according to the eligibility criteria. Specifically, the records have been included if: (a) they had been published between 2016 and 2019; (b) the main paper’s issue was “Occupational Welfare” in Italy and in Europe; (c) they were focused on innovative work-family life reconciliation policies and practices in Italy and Europe; (d) the papers focused on either policy or practices, therefore excluding theoretical and/or methodological papers. At the end of this process, 81 papers have been full text reviewed for final inclusion, of which 28 were included in this rapid review (Figure 1). The selected papers, as a part of the study’s materials, have been included and cited in the Results section as references, and are listed in the Appendix A of this article.

As for the expert interviews, they were chosen as a tool enabling the collection of information to gain an in-depth knowledge of the object of the study, through a rapid dialectic process between the investigating researchers and the involved “privileged witnesses” [23]. This methodology had been successfully implemented in previous international studies focused on innovations promoting active ageing and home care quality across Europe, such as Mobilising the Potential of Active Ageing in Europe (MoPAct) and Cost effectiveness and quality in long-term care (CEQUA), using country case studies as a basis to build a dataset suitable for international comparisons [24].

From October to December 2019, the first author conducted five face-to-face semi-structured expert interviews. The limited number of experts is a result of the fact that, in order to collect the most qualified and comprehensive opinions, only specialists were approached who had a recognized expertise in both LTC and OW related issues. Considering the only recently acquired relevance of LTC measures within the context of OW schemes, the Italian experts able to provide a qualified opinion on both topics are still very few. Moreover, we decided to use a mixed strategy of stakeholder/expert involvement [25], taking into account the relevance of their academic or professional profiles as reflected by their contribution to the national and sometimes even international debate concerning the investigated topic (Table 2). Therefore, the involvement of five experts still allowed to achieve an acceptable balance between different points of view and can be considered to be very close to a saturation level in terms of high-profile privileged witnesses in the investigated field.

In consideration of the dearth of specific literature concerning the Italian context, their profile is therefore based primarily upon practices and experiences conducted directly and reflected in the grey literature existing on this topic in Italy (e.g., as national reports), or via the knowledge acquired through specialised literature on national and international case studies. This expertise allows them to combine a systemic and national vision with a local, concrete experience. An invitation to participate in the study was sent by mail from the first author to each selected experts, indicating the general goals of the study, the list of main questions to be addressed, and the declaration of the anonymisation process to ensure the privacy of the interviewees. Five single interview meetings were scheduled according to the availability of the interviewee. Each interview lasted approximately 60 min and was recorded and transcribed according to Cohen’s [26] guidelines.

Table 3 shows the interview’s framework, identifying four main research goals: (a) description of the OW schemes in relation to the Italian LTC context; (b) identification of the various actors’ roles in it, paying particular attention to the informal care and MCW sectors; (c) identification of the core challenges faced by the OW schemes within the LTC system; (d) formulation of recommendations about potential innovations and future policy measures. One or more specific aims have been linked to each research goal (second column) and, finally, a set of questions has been built to respond in relation to those aims (last column).

Following the suggestions formulated within the theories developed by Titscher [27] and Kohlbacher [28], the interviews were analyzed by means of a qualitative content analysis based on a “multilevel interpretation” of data. This approach has been used to improve the pursuit of each of the investigated study goals, since it is based on an analysis that, starting from the research questions, identifies the responses to them by investigating the different “levels of context” implied by the investigated phenomenon. This allows researchers to unpack the relationship between underlying topics and the interaction between the different actors (i.e., what these say and do). In the specific case of this study, the expert interviews allowed, at a first level, to identify the opinions and/or suggestions concerning each of them. Sequentially, these views and suggestions were then grouped thematically (second level), in order to draw, in a third step, the final, overall framework of contents (third level). By doing so, it was possible to better understand the process of meaning construction with regard to the innovative role of OW within the specific context-sensitive conditions of Italian LTC, and to explain it by providing insights into its peculiarities.

Finally, to comply with the privacy rules aimed to assure anonymity for the experts interviewed, a set of abbreviations reflecting their professional profile has been used to sign their remarks (Table 2). No ethical approval was required for carrying out this kind of investigation according to the currently in force Italian legislation.

## 3. Results

In the following, the main results emerging from the study are reported. In order to facilitate the reader, they are illustrated according to the four research goals composing the framework depicted in Table 3 above. In particular, findings emerging from the experts’ comments and responses are integrated with the results suggested by the literature review, underling—where appropriate—any detected discrepancies between the two sources.

### 3.1. Occupational Welfare in the Italian LTC Context: Data Trends

In 2016 a set of innovations were introduced in the Italian welfare system by law No. 208/2015 labeled as “stability Law for 2016”. These innovations aimed to promote an increase in the supply of services to support employees with care responsibilities, including informal elder care provision [6,13]. The law allows three main measures: (a) tax incentives for companies that decide to grant welfare benefits for their employees; (b) the adoption of the voucher system to access services; and (c) the option to grant performance-related benefits in the form of welfare services to employees earning less than €50.000 per year. The basic idea is to integrate the public expenditure for the Italian LTC system, which in 2017 reached only 0.7% of GDP, i.e., one of the lowest among Organisation for Economic Co-operation and Development (OECD) countries (Table 4) with additional resources “activated” from the private sector. Moreover, the LTC budget in Italy is mainly absorbed by health care provided in hospitals (51%), leaving only a residual part available for home care (19%) and other forms of provision (30.1%). However, one aspect not covered by LTC spending is the care provided by MCWs and informal caregivers, which represent the bulk of LTC provision in Italy [3,4,29]. One of the most common ways for older adults to access care services is to pay them privately by using the cash allowances they receive from the State [15,16]. In 2017, more than 1.9 million older people received the carer allowance, for a total cost of 13.4 billion euros (equal to 0.8% of Italy’s GDP). This ensures implementation room for private providers, including MCWs [6,7]. A private propensity to spend in social support is confirmed by low public investment on social protection, since in Italy only 27% of GDP has been spent on social protection, one point below the European average (Table 4).

Despite the social protection expenditure highlighted Table 4, a recent tendency can be observed in Italy to pay an increasing attention to families and to social support, as shown by the fact that the Italian voluntary private social expenditure has doubled in ten years, soaring to 1.8% of GDP [29,30].

This trend is confirmed by the results of the rapid review, which underlines that in Italy the OW issue has gained a growing attention by policymakers, national and local stakeholders, and by the specialized literature in this field. In the last five years, different national studies have addressed OW and corporative welfare issues in Italy [31]. Similarly to what happened for other Italian welfare issues, these studies suggest an overall picture of a fragmented and multi-level OW system, based on different regulation acts and on the involvement of various stakeholders. The complexity of this system is based, on the one hand, on the fact that OW agreements can be broken down to two main levels:(a)National level of contractual OW agreements: these refer to the arrangements included in national collective agreements and/or adopted by large companies.(b)Local level of contractual OW agreements: these are mainly managed at the local level for specific sectors or applied by single or multi-enterprises.

In addition, single or multiple non-contractual agreements may be signed at enterprise level, based on the OW needs of the specific company.

As for the main stakeholders involved in the system, these are represented primarily by public institutions, bilateral bodies, trade unions and enterprises, all usually engaged in the co-design of OW agreements (Table 5). In this system, each stakeholder can have one or more operative functions in co-shaping and implementing the OW path. The specific home care measures included in OW agreements—as part of the whole spectrum of measures included in such agreements—operate in the same way.

Even though the lack of disaggregated data on OW experiences and agreements makes it difficult to detect precisely their impact in terms of home care measures, the data included in some sectorial studies gives us some clues on the ongoing trends in Italy. The Stability Law for 2016 repeals the voluntary nature of OW schemes in national collective agreements, thus making it now mandatory to include integrative OW schemes in the most recently renewed agreements. In the last years, moreover, the number of secondary level agreements offering social service provision has doubled, with an annual increase of around 30% [31].

Table 6 summarises the data highlighting the spreading of OW measures in contractual agreements undersigned at both first and secondary level, with over 167 thousand companies adopting OW schemes in 2018, for a potential population of up to 3.35 million workers. Barazetta and Santoni [32] highlight that this data means that in Italy OW schemes involve by now more than 11% of companies, and around 20% of employees.

Unfortunately, the measures promoting home care services are still thinly spread. While OW schemes supporting family care in general are included in 23% of secondary level OW agreements, those specifically aimed at supporting care provided by relatives are implemented in only 8% of them [34]. OW schemes are mostly concentrated in northern regions, where 69% of agreements have been signed, in contrast to only 13% in central regions and 2% in the South [32]. This trend is not surprising, considering that most companies, and especially the largest ones, are located in the northern part of Italy. Indeed, recent studies underline that most OW agreements are active in large companies, while only 26% of enterprises involved had less than 50 employees [32].

### 3.2. The Relationship between OW, LTC and Home Care

The relationship between OW, LTC and home care is strongly influenced by how OW has been implemented in Italy. *The main issue to be investigated, in this regard, should be: “What is the relevance of OW in welfare issues in Italy?”* (U3). The existing literature underlines that OW issues are still a relatively neglected area in the study of welfare state, despite its growing social and policy relevance. Indeed, OW policies support and influence transformations in the welfare state, for their ability to work in term of social protection, and for their potential to promote innovation [33]. The experts’ opinions on the reasons for this recent attention paid to OW policies come from the private nature of agreements between enterprises and employees. OW actions are born—and often still perceived—as individual choices made by large enterprises to support the productivity of their workers through additional benefits: *as such, it is considered as a sort of “internal” issue, and not one of public policy relevance* (U1). Despite over thirty years of active debate on the power of private welfare arrangements to integrate into the welfare state, *for a long time the supply of OW schemes has remained limited to an integrative and functional vision related to cash benefits for employees, as the sole direct beneficiaries of the company’s OW measures* (U2). The result has been the spread of a multi-pillar system that promoted an increased private provision, but perhaps also leading to a risk shift or to an “individualization of risk” [34]. *Unavoidably, these general widespread characteristics of OW policies have a limited impact in terms of implementation of specific OW home care schemes for frail older relatives* (U3). Despite these relatively negative opinions by the consulted experts about the past, these agree on the currently increasing relevance of OW schemes as a social protection measure supporting home-based care. The openness to home care support schemes contributes to the renewal of the role of OW: although home care support is only a recently introduced area of OW intervention, current evidence seems to suggest that OW can play a potentially “integrative role” with regard to the already existing care services and policies (PM1). Razetti and Maino [35] push the idea of an integrative function of all non-public welfare schemes, including OW, that can promote a mixed welfare system based on a fruitful collaboration between different sectors, services and stakeholders.

### 3.3. The Impact of OW Schemes on Italy’s Home Care System

OW schemes have been created to support the worker’s life. Not surprisingly, the most widely used schemes are based on insurance and conciliation measures, with their main direct users represented by the workers and their individual needs (CP1). In Italy, for many years OW schemes aimed at supporting LTC were made to tackle the risk of LTC needs among employees, so that most of the measures were constituted by long-term or health care insurances [36,37,38]. The Stability Law enlarge this spectrum, by specifically introducing measures earmarked for the care of employee’s older relatives. The grey literature analysis and the experts’ opinions allowed us to identify four different home care measures that may fall under this categorization within OW agreements: additional care leaves; cash benefits to support home care; vouchers to buy care services; and the direct provision of care by service providers. The additional care leave is used to extend the three days/month already permitted nationwide for care-related leave (Law No. 104/1992). The monetary measures—cash benefits and vouchers—aim to increase the ability to hire privately care workers to provide home-based care.

Finally, the provision of care services is aimed at supporting the fulfilment of care needs [9,39]. The main beneficiaries of these measures are therefore, in the first place, the employees’ disabled relatives, who become users and direct beneficiaries of specific OW measures assigned to their working relative (i.e., their family caregiver). Different measures can indirectly involve MCWs and providers of care services, too. In this regard, the target population of OW schemes is dramatically growing. In 2018, there were more than 859 thousand domestic workers (including care workers) formally employed in Italy, of which 71.4% were migrants (equal to 613 thousand) [18,19,20,40]. Moreover, it is estimated that 60% of MCWs are hired outside the formal economy, so that their real number has been estimated to reach a total of 1 million [1,14,41,42,43]. *De facto*, MCWs become indirect beneficiaries of OW schemes since cash benefit or voucher schemes are often specifically used as an additional means available to facilitate the home-based employment of MCWs. *This possibility is even explicitly included in some OW agreements, but they are too few to identify any impact on this sector. In any case, the positive effect of this measure should be to push for a legalisation of their job (U2).*

Recently, a national study on OW in Italy highlighted that more than 54% of employees would agree to exchange wage increases for OW measures [44]. *This is a changing company culture, and it takes time to be adopted* (PM1). Even though it is difficult to find data on the trends of home care schemes, we should note how widespread the involvement of private providers is: between 2015 and 2018, the number of companies that chose to commit providers to manage their OW schemes has grown six fold, involving a tripling in the number of workers as beneficiaries [43].

Concretely, the regulatory act allows workers an annual tax-free regime for expenditure in care or welfare measures of up to 2000 €/year. The employees can request the tax exemption regime for their total or partial expenditure. The services, vouchers and cash benefits received from OW schemes often contribute only partially to the total amount spent on home care services: the OW agreements often propose measures or cash benefits corresponding to 258 Euros [43].

The experts confirmed the accuracy of the abovementioned framework and underlined that the implementation of these schemes is still very limited and fragmented. *The measures specialized on home care provision have a low level of inclusion in OW agreements* and are also infrequently requested by workers (CP1).

One of the main reasons for this is their very recent introduction. The current legislation is pushing for an increasing role of OW schemes in supporting the work-life balance for workers that are informal carers, but it is too early to identify their impact, since *the cases are still too few, though positive trends can be detected* (U1). Actually, an overarching title to define OW measures supporting home care in Italy could be “Much ado about nothing”: *while the legislation is proposing to dedicate some OW measures to support home care, most companies continue to provide basic monetary schemes, because it is easier to manage them* (U3).

At the macro level, however, the OW has a strong impact on the reduction of tax revenue. Pavolini and Ascoli [45] underline in this regard that the Italian tax welfare and OW influence each other, reaching 3.1% of the total social expenditure [30]. *The main reason is the high monetarisation of OW schemes: supplementary health funds, supplementary pension funds and corporate welfare absorb most of the available resources (U3).*

### 3.4. The Room for OW Schemes to Promote Social Innovation within the Italian LTC

The experts’ opinions confirm that OW schemes applied to home care for older people can promote innovation in the Italian welfare system, including the LTC sector. *The OW framework applied to elder care needs brings forward an innovative and integrative welfare strategy* (PM1). The first innovation identified by the expert is the formal inclusion of LTC as action area in OW schemes. *Until 2016, in family support OW schemes the term “relative’s care” mainly meant “childcare”, so that measures had to be designed around the working parents, rather than around children that have to provide care to their ageing parents (CP1). The introduction of care for the elderly among the possible aims of OW agreements may help produce a cultural change on care and welfare issues (U1).* The inclusion of older relatives in the target population reinforces this assumption. *The parents of employees emerge as the new beneficiaries of measures* (CP1). Moreover, the “mandatory” character of OW agreements for companies after the Stability Law (which has enforced the adoption of such agreements, previously only optional) acquires an innovative function in terms of diffusion of home care services in OW programs. *If, until now, very few companies provided measures to support home care for the elderly, in coming years this number is bound to increase, through more agreements and more schemes specifically oriented to support home care* (U2). Moreover, the literature points out that the Stability Law adopted an innovative strategy to promote an enlarged use of welfare services—including home care—at the expense of the existing massive use of extra wage benefits [34]: *the law pushes for an empowerment of OW schemes in order to obtain a complete replacement of monetary benefits* (U2).

Despite this positive opinion on the potentially innovative push for the home care sector, the experts underlined that its concrete effects have yet to be detected. The framework designed is innovative, even though concrete results are still not visible (U3). *The short implementation time influences this lack of evidence, but to be very innovative and noticeable these measures must be included in larger strategies to improve the social protection system for workers and their families* (U1). *Over the last twenty years in Italy, the welfare system policies have been focused upon the collaboration between different stakeholders, so why shouldn’t we now adopt the same approach to support the renewal of OW schemes applied to family support?* (PM1). In particular, the consulted experts emphasized that the OW schemes ought to be considered as part of the larger social protection system, in order to enhance its integrative—rather than supplementary—potential function of public support.

### 3.5. Core Challenges and Policy Recommendations

The schemes applied to home care are the latest products introduced into the Italian OW system. The main challenge to their success is to move from theory to practice: *the regulatory system is in place, so now companies, trade unions and, not least, workers have to want to implement and use it* (CP1). This challenge is strongly related to the improvement of knowledge and information on OW schemes among stakeholders. Indeed, the experts agreed on the lack of clear information and knowledge, in particular by the direct users (workers and companies), on opportunities and advantages coming from OW policies and schemes. In these regards, they specifically recommended *the promotion of an informational and awareness campaign on OW schemes, aimed at companies but especially workers, to better use the existing possibilities to meet their care responsibilities* (PM1). The proposed informational campaign would contribute to spread home care schemes in OW agreements. A real challenge, in this respect, is *how to extend the room dedicated to home care schemes within OW agreements, by means of measures promoting specialised care services rather than cash or tax-free benefits* (U2). The enhanced provision of services, instead of monetary measures, is another challenge for OW schemes as well as for the LTC in Italy [46,47,48,49,50]: *a real revolution to this purpose would be accomplished by a greater presence and use of service measures, instead of the tax incentives that are the focus of current strategies* (U3).

The last identified challenge is the implementation of a long-term systemic strategy to enhance the relationships between OW schemes and the LTC. *In an ageing context and a familial care regime like the one existing in Italy, OW and care for older relatives must be connected: the work-life balance passes through these needs, so the challenge is to promote a long-time strategy. Remember it! (PM1)*. However, to become effective, the OW schemes applied to home care must be less fragmented and become more than merely individual experimental experiences: *the recommended strategy for OW schemes could be seen as integrative of traditional care services, in order to contribute to the local universal coverage of workers’ needs related to balancing working life and their care responsibilities (U2).*

## 4. Discussion

The illustrated findings show that OW schemes supporting home care are still far from being fully valued and integrated within the Italian LTC system, for a series of reasons.

In the first place, the residual attention traditionally granted to social protection and family support issues might have acted as a structural barrier preventing the adoption of measures like OW schemes to support home care provision [46]. Although the data on the Italian context confirms this assumption, the observed trend of an increase in the number and extension of OW agreements seems to suggest that, in a future perspective, these schemes might play a more substantial role in supporting home care policies in Italy.

The push received by the introduction of the national Solidarity Law in 2016 allowed for the start of a positive collaboration between two different areas of welfare—OW schemes and home care for the elderly—which until now had run independently of each other. This integrative process, however, has been strongly slowed down by the limited room given to both issues within national policy debates. Moreover, in Italy’s company culture OW has been so far seen more as a productivity bonus for employees, rather than as a social support opportunity to cover family care related needs. The integrative impact of OW schemes on formal home care provision—as identified by different experts—remains however to a large extent still only potential, as it needs to find more concrete forms of realization, in order to better respond to the challenges of an ageing population [2,48].

The illustrated results show, in this regard, that OW schemes supporting home care are made up of a highly complex system of measures, due in the first place to the fragmentation and multi-level governance existing in both Italian OW and LTC systems. As in other family-based care regimes (e.g., Spain and Israel), Italy’s multi-level care governance leaves indeed large room for the informal provision of care by younger relatives, and even more by women in working age [21,48,49].

From this point of view, the transformation from optional to mandatory OW agreements for companies that occurred in Italy after 2016 seems to reflect a broader intention by the government to start a process of strengthening and systematizing the support to informal care. Indeed, less than two years later, a draft national law containing a series of measures to support informal caregivers was presented. Taking inspiration from legislation in other countries (e.g., the Spanish national reform on LTC), this law included a specific definition of familial caregivers and identified tax, social security and work inclusion measures for them [2,8,50]. Due to the end of the legislative period, the law was not adopted, but it was a clear sign of the increasing attention paid by the Italian society and its national political representatives to informal care issues and to the need of providing more systemic responses to unmet LTC needs. Within this context, the consulted experts highlighted the socially innovative and system-integrative role played by OW schemes applied to the LTC sector, also in terms of facilitating the work-life balance of working family carers, thus contributing to the improvement of the quality of life of large segments of the population [51,52]. This is partly connected also to the multi-stakeholder collaboration in the delivery and design of LTC strategies established at local level by several OW schemes, which could further increase their innovative character. A sound example of this kind is proposed by Maino and Razzetti’s “open scenario for local welfare”, which builds on the joint contribution of public institutions and services, NGOs, confessional organizations, bank foundations, private providers and companies, as a network of different but integrated promoters of OW schemes, including home care [52].

Less than five years since the first introduction of OW schemes in 2016, it may still be too soon to evaluate properly the impact of these measures, due to their low—albeit increasing—uptake rates by both companies and employees. This is not surprising, in the context of a gender-disadvantaged labor market like the Italian one, with 38% of female workers that report to have caring responsibilities, compared to just 12% among men [53]. While therefore OW schemes have the potential of being helpful to improve the work-life balance of informal caregivers, further research is however needed to better understand the existing barriers preventing a wider implementation of these schemes. In this regard, Natali et al. [47] have found that access to occupational benefits is not evenly distributed among all socio-demographic groups and workers, with women and self-employed workers being less likely to be able to access benefits. This is of particular concern in Italy, due to the high degree of self-employed workers. Moreover, the preference traditionally expressed by Italian citizens towards monetary compensations—rather than in-kind reconciliation services—is deeply rooted in the country’s familial-based welfare culture [54,55,56]. This contributes to explain why the increase of OW schemes in the home care sector, while it certainly has an overall potentially beneficial impact, may represent at the same time also a possible cause for concern, if it ends up with leading, as it seems, to an increased recruitment of MCWs outside the formal economy [54]. Thus strengthening the pattern, already existing, of the State cash allowances used by families to hire care workers on an undeclared basis, contributing to perpetuate a situation of precarious employment conditions for MCWs, on the one hand, and of a low quality in home-based care, on the other hand. Efforts to tackle this challenge have been so far primarily limited to the adoption of regularization campaigns for undocumented MCWs by the Italian government (the last one going back to 2009, determining a 20 per cent increase in the number of legal MCWs in Italy) [57]; hence, given the ad hoc, temporary nature of these initiatives, these cannot offer a long-term systematic solution to this issue.

Therefore, while OW schemes for home care have been as yet a rather residual experience within the Italian context, they have a massive potential for further increase in the near future. According to recent estimates, well 760 thousand workers could choose to receive it, if the already activated OW agreements would include home care schemes, and at least 50% would be using them [58]. This scenario seems realistic, taking into consideration that, in 2018, almost 400 thousand workers required daily job leaves for family care purposes, and that the requests have increased by 30% in a few years. These results show at the same time, however, that in order to make sure that OW schemes in home care can go beyond the level of a sort of “start-up” experience, robust and thought-through communication and information actions are needed to make them a more significant component of the Italian LTC system. All experts’ recommendations suggest, in this regard—and in doing so they seem to be in line to what the available literature suggests [32,44]—to move towards a consolidation and extension of these measures, by means of a more synergetic strategy of integration between public provision of services and cash-for-care allowances, on the one hand, and OW schemes promoting a more market-based provision, on the other hand.

## 5. Conclusions

This study aimed to describe how the newly introduced OW schemes might innovate existing home care arrangements in Italy. A rapid review of the literature alongside expert interviews allowed us to describe the complex Italian OW schemes and identify the positive implications for their application to the local LTC systems. The high fragmentation and complexity of both OW and LTC work as barriers to an extension of OW schemes to the field of home care support. Their still relatively short implementation time does not allow for the detection of quantitative precise data on their concrete effects for workers (including informal caregivers and MCWs). However, the qualitative information collected via experts identified a series of innovative elements in the OW schemes applied to Italy’s home care sector. Among them, they seem to promote social innovation for their potential ability to: (a) better fulfil unmet conciliation and care needs; (b) improve quality of life for beneficiaries (family carers and their older relatives); (c) be included into a larger multi-stakeholder and multi-level welfare strategy.

Concretely, the extension of OW schemes in the home care sector may lead to a series of enhancements at different levels. At the *macro level*, it might facilitate an improvement in the level of coordination and integration among different welfare policies, thanks to a more systematic dialogue between the two areas of welfare, which are usually separated. At the *meso level*, the growth of involvement by companies in social care responsibility could push them to be more strongly involved in the networks of local and national stakeholders, and a positive spin-off of this participation may be the building of new collaborations between private and public stakeholders. Moreover, companies may be prompted to think about the promotion of new useful solutions to support their workers in addressing their family care needs without loss of productivity at work and, simultaneously, to improve the organisational well-being. As for the *micro level*, the quality of care provided at home may be improved by a better implementation of OW schemes in the home care sector, by integrating different measures, such as “for instance” by expanding care leaves for informal carers, providing more direct contact hours, or ensuring a better monitoring and improved working conditions and care quality for MCWs. In this regard, useful measures might be represented by a facilitated used of vouchers, fiscal benefits for expenses borne to hire MCWs, or traceable cash benefits for an easier regulation of MCWs’ recruitment and employment conditions. No matter which level is referred to, OW schemes in home care would in any case need, to be implemented more extensively, ad hoc communication campaigns to inform a largely unaware audience.

Finally, some limitations should be considered in interpreting the results of this study. A first, essential drawback is represented by the choice of relying on a rapid review, rather than on a systematic review, to scrutinize the literature on the investigated topic. While this has been an explicit decision to reach potentially translational findings within a shorter time horizon, it might have reduced the depth and extent of information acquired. Secondly, the rapid review has been realised including Google and google scholar. While their use has been necessary to provide a larger search considering the low spread of specialised literature on the investigated topic, these search engines ensure only a limited level of precision, because they present data based on citations and other factors. Third, the lack of disaggregated data does not allow for the possibility of analyzing home care provision independently of OW schemes. Moreover, the focus on Italy as a single country case study, with no comparative data to refer to, makes it difficult to draw generalized conclusions for the international context. Finally, the mixed strategy used to involve the five consulted experts, while allowing to achieve a first analysis of the Italian case in light of the existing literature on this issue, still represents a limited perspective, since different local experiences require specific investigations and the involvement of a larger number of stakeholders.

Despite these limitations, the findings provided by this study represent, to our knowledge, the first attempt to provide to a non-Italian audience of readers an in-depth examination of an innovative—and cross-nationally potentially transferable—experience to address a politically very relevant issue. Further studies are however urgently needed in this regard, which should possibly count on a systematic analysis of the literature, a more extensive primary data collection, and the involvement of a larger sample of stakeholders representing a wider variety of perspectives, also in terms of regional differentiation.

## Figures and Tables

**Figure 1 ijerph-17-05511-f001:**
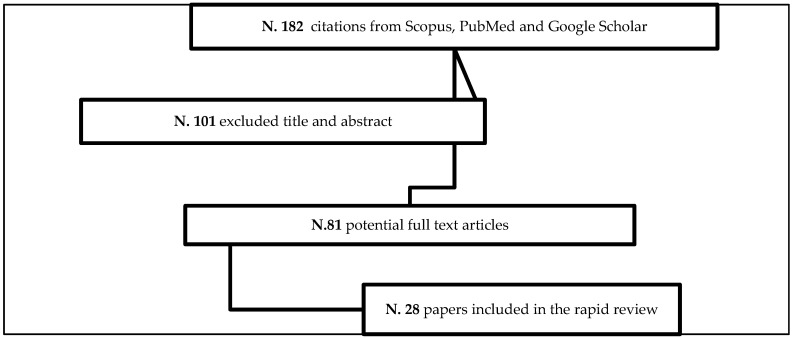
Flowchart of the rapid review process.

**Table 1 ijerph-17-05511-t001:** Italian care needs in a comparative perspective: ageing trends, share of population with care responsibility, and share of informal carers across selected European countries.

Country	% of Population Aged 65+		% of Population Aged 18–64 with Care Responsibility for Incapacitated Relatives (2018)	Informal Carers as a % of Total Population (2016)					
	2018	Difference (2018/2009)		Total	Male	Female	18–35	35–64	65+
Greece	21.8	3.0	10.1	34.0	29.0	39.0	35.0	35.0	34.0
Netherlands	18.9	3.9	10.0	18.0	13.0	23.0	10.0	23.0	17.0
Italy	22.6	2.3	7.7	17.0	16.0	19.0	12.0	20.0	18.0
Spain	19.2	2.6	6.9	16.0	13.0	19.0	14.0	19.0	11.0
Poland	17.1	3.6	6.7	20.0	18.0	21.0	7.0	26.0	20.0
Portugal	21.5	3.5	6.1	13.0	9.0	17.0	8.0	17.0	10.0
Ireland	13.8	2.9	5.7	10.0	9.0	11.0	6.0	13.0	8.0
Malta	18.8	4.6	5.6	26.0	25.0	27.0	19.0	33.0	21.0
EU	19.7	2.4	5.5	17.0	15.0	20.0	13.0	20.0	17.0
France	19.7	3.2	5.3	16.0	13.0	18.0	11.0	18.0	14.0
Belgium	18.7	1.6	4.9	30.0	23.0	36.0	25.0	33.0	30.0
Cyprus	15.9	3.4	4.9	15.0	10.0	20.0	11.0	18.0	13.0
Hungary	18.9	2.5	4.5	18.0	17.0	18.0	8.0	25.0	13.0
Austria	18.7	1.3	4.4	10.0	8.0	12.0	4.0	11.0	15.0
Finland	21.4	4.7	3.7	13.0	10.0	16.0	6.0	16.0	13.0
Sweden	19.8	2.0	3.6	12.0	10.0	15.0	12.0	14.0	10.0
Germany	21.4	1.0	2.9	23.0	20.0	26.0	12.0	28.0	24.0
Romania	18.2	2.1	2.7	9.0	8.0	10.0	5.0	11.0	9.0

Sources: Eurostat, 2019, EQLS 2016.

**Table 2 ijerph-17-05511-t002:** Experts and stakeholders involved in the study: typologies, number, and abbreviations.

Typologies of Involved Stakeholders	No. of Interviewed Experts	Abbreviations Used in the Text
Universities	3	U1; U2; U3
National research institution and policy maker	1	PM1
Care providers association	1	CP1
Total	5	

**Table 3 ijerph-17-05511-t003:** Interview framework: research goals, aims and questions.

Research Goals	Aims	Questions
Description of the Occupational Welfare schemes in relation to the Italian Long-Term Care/home care context	To define the main OW schemes applied in Italy in relation to home care	How would you define the OW schemes in Italy? Which OW schemes in Italy are related to LTC provided in the home? How would you define the relationship between OW schemes and LTC provided in based home care?
The impact on various actors’ roles	To understand stakeholder networks and the related roles of each To understand the OW schemes impact on home care	What is the role of the different types of stakeholders involved in OW schemes? What is the impact of OW schemes for the main Italian home care stakeholders (informal carers and Migrant care Workers - MCWs)?
Identification of core challenges	To define the challenges of OW schemes applied on home care	What are the main challenges currently facing the OW schemes supporting LTC at home, in your opinion? How can each of these challenges be transformed to allow for the opportunity for innovation?
Formulation of recommendations on innovations and policy	To define what innovative strategies should be used and to define what policymakers can do to support OW schemes supporting home care	What could be some innovative strategies to support and improve OW schemes supporting home care provision in Italy? What main recommendations do you have for policymakers?

OW is Occupational Welfare.

**Table 4 ijerph-17-05511-t004:** Social protection and LTC expenditure in Italy and in the international context.

Social Protection Expenditure (SPE) 2016	EU	Italy
SPE Total/GDP (%)	28.4	27.1
of which for:	=100.0	=100.0
Health	29.5	23.1
Pensions for retired adults	45.6	57.8
Disability	7.4	5.8
Family support	8.7	6.3
Unemployment support	4.6	5.8
Social exclusion support	4.2	1.0
LTC Expenditure (2017)	OECD *	Italy
Total % of GDP	1.7	0.7
of which for:	=100.0	=100.0
Inpatient long-term care	62.3	51.0
Home-based long-term care	33.2	19.0
Other	4.5	30.1

* Organisation for Economic Co-operation and Development. Source: elaboration by authors based on [28,29].

**Table 5 ijerph-17-05511-t005:** The Occupational Welfare system in the Italian LTC context: main role played by different stakeholders involved in co-design and implementation.

Stakeholders	Role Played by in OW Path			
	Regulation	Buyer for Employees	Seller	Provider
Public institutions	X	X		X
Enterprises	X	X		
Bilateral bodies	X	X		
Trade unions	X	X		
Trade associations	X	X	X	
Providers (profit/NGOs *)	X	X	X	X

* Non-governmental organizations; Source: elaboration by authors based on [31,32,33,34].

**Table 6 ijerph-17-05511-t006:** Occupational Welfare in Italy: Enterprises and workers involved at the first and second level of contractual agreement.

Contractual Agreements	Enterprises	Workers Beneficiaries
First level (2018)	166,011	2,432,098
Secondary level (2017)	1078	928,260
Total	167,089	3,350,358

Source: [31].

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
