# Peer review of "Between Migrant Care Work and New Occupational Welfare Tools: Changing Home Care Arrangements in Italy"

_ijerph, 2020, doi:10.3390/ijerph17155511_

Round 1

Reviewer 1 Report

An interesting paper that covers an important topic. However, I have a number of methodological, analytical and substantive concerns, and, some suggestions for how this paper could be revised.

Firstly, there is far too little information regarding the 28 articles you have identified for the "rapid literature review." You have identified the keywords used to search the databases but have not identified the articles settled on for your research. Shouldn't these be included as an appendix to your article or are these in your References? Moreover, conducting a standard literature review for a scientific article hardly qualifies as a raw data gathering exercise on which to base your analysis and scientific findings per se. Presenting a fundamental requirement for conducting research on an issue/concern as a virtue, "a rapid dialectical process," does not make it so. 

Secondly, you conducted five interviews with experts or so-called "privileged witnesses." Three were at universities, one at a research institute, and another at a caregiver association. Why were these five persons selected other than your qualification that they are experts? In addition, what can meaningfully be gleaned from interviewing only five people, three of whom are based at universities, rather than those who are practitioners in this field? I question the reliability of your findings based on a mere five "experts." The results of these interviews may be interesting and suggestive, but, they can hardly be used to base any solid generalizable findings base on so few "elite interviews."

Moreover, it would have been good to provide a complete list of your interview questions, along with how the interviews were conducted, how long they took place, how the data was captured, etc. Further, did you obtain research ethics clearance to conduct these interviews? If so, this should be included and you should explain how it was obtained.

The article purports to tackle the difficult question of how the Occupational Welfare (OW) schemes, following the 2016 Stability Law, might innovate existing LTC care arrangements in Italy. However, I did not detect any major findings in this regard other than it is too early to detect any evidence thereof.

Nor did I detect any major policy recommendations with respect to the OW schemes and caregivers in Italy. The Conclusions mention the potential to better fulfil unmet conciliation and care needs and to publicize OW through communications and information campaigns. One would have hoped for more substantive recommendations, especially, as it may effect MCWs, the predominant personal carers in both the formal and informal economy.

I detected a number of sloppy errors in the paper and list some of these below: 

At lines,

23 the abbreviation for Migrant Care Worker (MCW) is missing in the abstract and is used throughout the article.

27 likewise, but in the inverse, the LTC acroynm is found in the abstract without its attendant full reference, long term care.

42 delete space between (including ...

53, Table 1, does not show Switzerland, Iceland or Luxenbourg, yet, they are referenced in the commentary.

106 the first quotation marks are missing before Occupational welfare"

Figure 1, potential full articles not potetial

126 is num not number?

At the top of page 5 WO should be OW

160 remove space at 27%

173 WO is OW

239 LTC not 'the' LTC

263 is Censis not Census?

270 euro/year should be Euros/year?

274 abovementioned is above mentioned

315 renew should be renewal

355 OW welfare should just be OW

523 and 536 there are spacing issues

A further deep read of the text seems to be in order to eliminate these types of obvious errors.

Author Response

Dear Reviewer, many thanks for your in-depth reading and your useful suggestions to improve the paper. In the following, you will find specific answers to each of your suggestions/questions. 

#

Section

 Reviewer’s suggestions and/or questions

Authors’ answers

1.                  

Methods

Firstly, there is far too little information regarding the 28 articles you have identified for the "rapid literature review”. You have identified the keywords used to search the databases but have not identified the articles settled on for your research. Shouldn't these be included as an appendix to your article or are these in your References?

We prepared and added a list of all papers resulting from the rapid literature review as an Appendix to the article.

2.                  

Moreover, conducting a standard literature review for a scientific article hardly qualifies as a raw data gathering exercise on which to base your analysis and scientific findings per se. Presenting a fundamental requirement for conducting research on an issue/concern as a virtue, "a rapid dialectical process," does not make it so. 

Within the methodological section we tried to explain in a more detailed way the reasons – including time – which have led us to choose the proposed mixed method of data collection instead of other, more rigorous approaches. As we agree with the reviewer that this issue remains a basic limitation of this study, we have mentioned it as additional drawback to be overcome by future studies in the final section of the article.

3.                  

Secondly, you conducted five interviews with experts or so-called "privileged witnesses." Three were at universities, one at a research institute, and another at a caregiver association. Why were these five persons selected other than your qualification that they are experts? In addition, what can meaningfully be gleaned from interviewing only five people, three of whom are based at universities, rather than those who are practitioners in this field? I question the reliability of your findings based on a mere five "experts." The results of these interviews may be interesting and suggestive, but, they can hardly be used to base any solid generalizable findings base on so few "elite interviews."

We have specified in a more analytical way the process that has led to identification and selection of the ideal profile of the experts chosen for the interviews. The explanation provided hopefully clarifies why it was so difficult to find a larger number of experts possessing the specific expertise required to provide a qualified opinion on the investigated topic. Again, we agree that this might also represent a limitation of the illustrated study, and included it as such in the corresponding section at the and of the article.   

4.                  

Moreover, it would have been good to provide a complete list of your interview questions, along with how the interviews were conducted, how long they took place, how the data was captured, Further, did you obtain research ethics clearance to conduct these interviews? If so, this should be included and you should explain how it was obtained etc.

Table 3 contains the complete list of the broad open-ended questions we used as a framework to “nudge” experts’ views on the investigated topic. All other requested details on the interview process have been added in the text, including reference to the ethical approval.

5.                  

Findings and Conclusions

The article purports to tackle the difficult question of how the Occupational Welfare (OW) schemes, following the 2016 Stability Law, might innovate existing LTC care arrangements in Italy. However, I did not detect any major findings in this regard other than it is too early to detect any evidence thereof. Nor did I detect any major policy recommendations with respect to the OW schemes and caregivers in Italy. The Conclusions mention the potential to better fulfil unmet conciliation and care needs and to publicize OW through communications and information campaigns. One would have hoped for more substantive recommendations, especially, as it may effect MCWs, the predominant personal carers in both the formal and informal economy.

The conclusion section has been integrated by additional sentences concerning possible policy implication and recommendations emerging from the study.

Additional remarks

I detected a number of sloppy errors in the paper and list some of these below.

We have corrected all errors indicated, and reviewed the whole text to remove additional mistakes and redundant sentences.

6.                  

The abbreviation for Migrant Care Worker (MCW) is missing in the abstract and is used throughout the article. Likewise, but in the inverse, the LTC acronym is found in the abstract without its attendant full reference, long term care.

Done: the acronym MCW has been added.

7.                  

42 delete space between (including ...

Done

8.                  

53 Table 1, does not show Switzerland, Iceland or Luxenbourg, yet, they are referenced in the commentary.

Done: the commentary includes now only countries listed in Table 1.

9.                  

106 the first quotation marks are missing before Occupational welfare"

Done

10.                

Figure 1, potential full articles not potetial

Done

11.                

126 is num not number?

Done

12.                

At the top of page 5 WO should be OW

Done

13.                

160 remove space at 27%

Done

14.                

173 WO is OW

Done

15.                

239 LTC not 'the' LTC

Done

16.                

263 is Censis not Census?

Done: “Censis” was right, but we removed it from the text, reformulating the sentence.

17.                

270 euro/year should be Euros/year?

Done

18.                

274 abovementioned is above mentioned

Done

19.                

315 renew should be renewal

Done

20.                

355 OW welfare should just be OW

Done

21.                

523 and 536 there are spacing issues

Done.

22.                

A further deep read of the text seems to be in order to eliminate these types of obvious errors

A deep read of the text has been completed.

Reviewer 2 Report

Thank you very much for the opportunity to review this work. The authors have proposed an interesting investigation. However, in order to improve their presentation and quality, I think they should make some changes.

The authors have done a good job in the introduction section by contextualizing the study, but this section needs to be completed. Regarding originality, some points could be developed in the introduction section. Comment. Please, clarify why your paper is important. What are you going to discover? Why is this topic important? Describe knowledge gaps.

The methodology used must be explained in more detail. Specifically, the authors must clearly state what they have relied on to design the questionnaires. Above all, they should point out if previous studies in this field have used similar methodologies. It is important to justify why they have used this methodology, to point out studies in which it has been used and to expose which methodologies have been used in close studies.

Limited findings and discussion section relating to the literatura.

Is there a relationship between the literature review and the results section of this paper? 

Are there any discrepancies on result?

The conclusions need to be less routine and more interesting and relevant to your findings.

Practical implications?? There is nothing here to inspire future research or implications for practice

Author Response

Dear Reviewer, many thanks for your in-depth reading and your useful suggestions to improve the paper. In the following, you will find specific answers to each of your suggestions/questions. 

General comments

Thank you very much for the opportunity to review this work. The authors have proposed an interesting investigation. However, in order to improve their presentation and quality, I think they should make some changes.

#

Section

Reviewer’s suggestions and/or questions

Authors’ answers

1.                  

Introduction

The authors have done a good job in the introduction section by contextualizing the study, but this section needs to be completed. Regarding originality, some points could be developed in the introduction section. Comment. Please, clarify why your paper is important. What are you going to discover? Why is this topic important? Describe knowledge gaps.

The introduction has been revised and integrated in order to highlight more clearly the importance of the investigated topic, and the contribution that the study can provide to address it.

2.                  

Methodology

The methodology used must be explained in more detail. Specifically, the authors must clearly state what they have relied on to design the questionnaires. Above all, they should point out if previous studies in this field have used similar methodologies. It is important to justify why they have used this methodology, to point out studies in which it has been used and to expose which methodologies have been used in close studies.

The methoology has been extensively revised and extended, indicating the previous studies using the same approach, and explaining in more detail the methodological steps followed to achieve the study’s aims.

3.                  

Discussion

Limited findings and discussion section relating to the literature. Is there a relationship between the literature review and the results section of this paper? Are there any discrepancies on result?

The discussion has been thoroughly revised and integrated with additional comments and direct references to the literature.

4.                  

Conclusions

The conclusions need to be less routine and more interesting and relevant to your findings.

The conclusions have been extended, and include now also a more developed section highlighting the limitations of the study (moved from the discussion).

5.                  

Practical implications?? There is nothing here to inspire future research or implications for practice

Additional remarks in terms of implications for policy and practice at micro, meso and macro levels have been included in the revised version of the article.

Round 2

Reviewer 1 Report

Thank you for addressing all my concerns as outlined in my initial review. 

I still think that there is some room for improvement of the introduction so that it ties in better with your overall conclusions.

The findings of your study could have related more to your quick literature review and the list of questions posed to the five expert respondents interviewed. These are covered in the Discussion section of the paper but not in a structured or thematic way, at least as far as I could tell.

And, you may wish to consider the following minor corrections and/or changes:

Line 42  - de facto should probably be in italics, and, probably at line 308 as well;

Line 55  - the sentence ending with "too" seems awkward (is a comma required immediately before it?);

     114  - there should be commas around "in an agile manner";

     232  - Table 5, the spacing is off for the word "institution" which makes it run over to another line.

     295  - correct the split infinitive to "to hire privately";

     435  - correct the split infinitive to "to evaluate properly";

     453  - insert a space at "determining a";

     493  - put commas around "for instance". 

Author Response

Dear Reviewer, many thanks again for your interest in our paper and your support in improving it through your suggestions. Below our additional work to address them. Best regards, the authors. 

#

Section

 Reviewer’s suggestions and/or questions

Authors’ answers

1.                 

Introduction

Thank you for addressing all my concerns as outlined in my initial review. I still think that there is some room for improvement of the introduction so that it ties in better with your overall conclusions.

To better link the introduction with the conclusion, some additional sentences have been included in the introduction, connecting in a more systematic way the two sections of the article.

2.                 

Results

The findings of your study could have related more to your quick literature review and the list of questions posed to the five expert respondents interviewed. These are covered in the Discussion section of the paper but not in a structured or thematic way, at least as far as I could tell.

In order to facilitate the reader, an introductory sentence has been inserted at the beginning of the Results section of the paper, explaining how findings have been reported in the same (thus making clear that, instead of following each single question of the study’s topic guide, the corresponding “research goals” have been followed to present the results).

The discussion section has been also partly revised, highlighting the most important messages in bold, in order to deliver them in a more immediate and structured way to the reader.

3.                 

And, you may wish to consider the following minor corrections and/or changes:

Line 42 - de facto should probably be in italics, and, probably at line 308 as well;

Done

Line 55 - the sentence ending with "too" seems awkward (is a comma required immediately before it?);

Done

114 - there should be commas around "in an agile manner";

Done

232 - Table 5, the spacing is off for the word "institution" which makes it run over to another line.

Done

295 - correct the split infinitive to "to hire privately";

Done

435  - correct the split infinitive to "to evaluate properly";

Done

453  - insert a space at "determining a";

Done

493  - put commas around "for instance". 

Done

Reviewer 2 Report

I think the authors have done a good job, the article is now more clearly presented and the content is more developed. In my opinion, this version is suitable for publication.

Author Response

Many thanks for your positive comment.